# Natural Compounds for Counteracting Nonalcoholic Fatty Liver Disease (NAFLD): Advantages and Limitations of the Suggested Candidates

**DOI:** 10.3390/ijms23052764

**Published:** 2022-03-02

**Authors:** Noel Salvoza, Pablo J. Giraudi, Claudio Tiribelli, Natalia Rosso

**Affiliations:** 1Fondazione Italiana Fegato—ONLUS, Area Science Park Basovizza, SS14 km 163.5, 34149 Trieste, Italy; noel.salvoza@fegato.it (N.S.); pablo.giraudi@fegato.it (P.J.G.); 2Philippine Council for Health Research and Development, DOST Compound, Bicutan, Taguig 1631, Philippines

**Keywords:** coffee, caffeine, caffeic acid, chicoric acid, tormentic acid, verbascoside, silymarin, NASH, NAFLD

## Abstract

The booming prevalence of nonalcoholic fatty liver disease (NAFLD) in adults and children will threaten the health system in the upcoming years. The “multiple hit” hypothesis is the currently accepted explanation of the complex etiology and pathophysiology of the disease. Some of the critical pathological events associated with the development of NAFLD are insulin resistance, steatosis, oxidative stress, inflammation, and fibrosis. Hence, attenuating these events may help prevent or delay the progression of NAFLD. Despite an increasing understanding of the mechanisms involved in NAFLD, no approved standard pharmacological treatment is available. The only currently recommended alternative relies on lifestyle modifications, including diet and physical activity. However, the lack of compliance is still hampering this approach. Thus, there is an evident need to characterize new therapeutic alternatives. Studies of food bioactive compounds became an attractive approach to overcome the reticence toward lifestyle changes. The present study aimed to review some of the reported compounds with beneficial properties in NAFLD; namely, coffee (and its components), tormentic acid, verbascoside, and silymarin. We provide details about their protective effects, their mechanism of action in ameliorating the critical pathological events involved in NAFLD, and their clinical applications.

## 1. Introduction

Nonalcoholic fatty liver disease (NAFLD) is strongly associated with clinical conditions such as overweight or obesity, type 2 diabetes mellitus (T2DM), hypertension, hypertriglyceridemia, and low (high-density lipoprotein) HDL cholesterol, all of which constitute the essential elements in the spectrum of the metabolic syndrome (MS). For these reasons, NAFLD is now considered as the hepatic manifestation of MS [1]. The booming rates of obesity and other MS elements contribute to the increasing worldwide prevalence of NAFLD in adults and children. Therefore, NAFLD incidence in children and adolescents represents a major public health threat in the upcoming years [2,3]. 

NAFLD is an umbrella term that comprises a wide spectrum of disease and commonly represented by two phenotypes, namely nonalcoholic fatty liver (NAFL) and non-alcoholic steatohepatitis (NASH). While NAFL is considered a relatively benign and reversible accumulation of lipids within the hepatocytes (simple steatosis), NASH is considered a more severe and often progressive form of the disease [4,5]. NASH tends to progress to fibrosis and cirrhosis, and eventually to hepatocellular carcinoma (HCC) [6].

The “multiple hit” hypothesis, a more complex and global theory to explain NAFLD pathogenesis, is widely accepted [7]. This hypothesis takes a wider look of the multitude of factors that affect NAFLD development, from the intrahepatic lipid accumulation to the involvement of “nonhepatic players” such as adipose tissue and gut-related mechanisms. Moreover, genetic and epigenetic factors also play a role in the disease manifestation [7,8,9]. Nevertheless, the onset of the disease is still represented by the accumulation of fat in the liver [9]. Intrahepatic lipid accumulation leads to some critical pathological events associated with NAFLD development, such as insulin resistance, oxidative stress, inflammation, and fibrosis [5]. These events have been investigated as potential therapeutic targets of NAFLD.

There is no consensus concerning an effective pharmacological treatment for NAFLD, and the only currently recommended alternative relies on lifestyle modifications [9] (diet [10,11] and physical activity [12,13,14]). However, the lack of compliance is still hampering this easy and cheap approach. The available drug interventions are mainly based on the association of several compounds as an attempt to reverse the comorbidities of the metabolic syndrome. Unfortunately, to date, any proposed drugs have not provided solid results [15]. Thus, there is an evident need for the characterization of new therapeutic alternatives. 

Studies of food bioactive compounds became an attractive approach to overcome the reticence toward lifestyle changes. Several natural compounds have been shown to exert beneficial effect(s) in the cellular mechanisms involved in the onset and progression of different diseases. These pieces of evidence represent a promising strategy for NALFD, whose pathogenesis is multifactorial and complex. Despite the reported results on experimental models, the translation to the clinical setting has been disappointing due to the variations in the effective dosage, bioavailability, duration of the treatment, differences in the purity of the compound, and lack of standardization [16,17,18]. 

The present study aimed to review some of the reported compounds with beneficial properties in NAFLD; namely, coffee (and its components), tormentic acid, verbascoside, and silymarin. We provide details about their protective effects, their mechanism of action in ameliorating the critical pathological events involved in NAFLD, and their clinical applications.

## 2. Coffee 

Coffee prepared from the seeds of the coffee plant (genus *Coffea*) is one of the most consumed beverages in the world. Almost 30 years ago, an association between coffee caffeine consumption and a decreased risk of liver disease was established [19]. Over 100 compounds have been identified in coffee extracts, and the synergistic effects of several compounds could contribute to the hepatoprotective health benefits reported worldwide by the scientific community [20,21]. In this section, we review the most recent literature, summarizing the effects observed on NAFLD of caffeine and two phenolic compounds: caffeic acid (CA), widely consumed in the human diet and present in most plants (including coffee); and chicoric acid (ChiA), a derivative of caffeic and tartaric acids, isolated for the first time from *Cichorium intybus*, but occurring in several plants.

### 2.1. Caffeine

The scientific evidence on coffee consumption and its effects on NAFLD reported in the last 10 years are summarized in Table 1. Here, we will review the main findings described in those systematic reviews. Saab S. et al. [21] reviewed all the studies published from 1986 to 2012, covering both observational and case-controlled studies. They explored the interaction between coffee consumption in liver-associated tests, viral hepatitis, NAFLD, cirrhosis, and hepatocellular carcinoma. As regards NAFLD, extensive evidence has highlighted the beneficial effect of coffee consumption and caffeine (IUPAC name: 1,3,7-trimethylpurine-2,6-dione)—the most abundant component in coffee beans—as the main compound responsible for the decreased risk of disease development [21,22,23].

Moreover, two of the five studies analyzed by Saab also argued that coffee decreased the risk for the worst stages of NAFLD, meaning fibrosis alone or fibrosis and NASH [20]. However, controversies remain regarding whether espresso or filtered coffee has the same beneficial effects on NAFLD [24]. Chen S. and colleagues [25] were the first authors who published a systematic review focusing on the relationship between coffee consumption and NAFLD. They enumerated and discussed the primary mechanisms for which coffee has been implicated in NAFLD severity reduction, highlighting that the hepatoprotective effects of coffee can be related more to polyphenols than caffeine. In short, evidence on polyphenols and their protective properties may be manifested through several mechanisms, including antioxidant, anti-inflammatory, and antifibrotic pathways; modulation of energy metabolism; reduced insulin resistance; and reduced severity of diabetes [25]. Among the most recent meta-analyses, the study of Hayat M. et al. [26]. summarized in a first analysis the evidence of the effects of coffee intake on NAFLD between drinkers versus nondrinkers. In a second analysis, they compared the risk of liver fibrosis development in the same drinking setup of NAFLD patients. The results of the meta-analysis, which included 11 studies, revealed a 23% decreased risk of development of NAFLD in subjects that drank coffee regularly. Moreover, it showed that those subjects with diagnosed NAFLD that consumed coffee regularly had a 32% reduced risk of developing fibrosis.

The main limitations of these meta-analysis studies were: (a) an unstandardized definition of coffee consumption among the included articles; (b) incomplete relevant information on coffee intake (such as type of coffee used, brewing method, coffee components, caffeinated or decaffeinated, consumption of other caffeinated beverages, etc.); (c) generally, meta-analyses included only observational studies showing associations, and not a causal relationship of coffee as a hepatoprotective agent; and (d) studies were biased (mainly by gender imbalance, cofounders, etc.). 

Considering the limitations mentioned before, further studies using a broad range of experimental systems with a well-defined dosage of coffee components will be necessary to establish the causal factors underlying coffee hepatoprotection. From the literature retrieved on this review, only two studies combined in vivo and in vitro models to explain the molecular mechanisms and pathways involved in the protective observations by caffeine at the hepatic level. Zhang S-J. et al. [27] investigated the effects and molecular mechanisms in which caffeine acts against hepatic steatosis in a high-energy diet model (HED mice). In this mice model, liver steatosis induced by the diet was diagnosed histologically, and several biochemical parameters were analyzed after the gavage administration of caffeine (10 or 20 mg/kg). Under this experimental setup, researchers found that caffeine significantly decreased the mass of fat tissues, lipidemia, and transaminases blood levels.

The study by Fang C. et al. [28] used a high-fat-diet (HFD) mouse model to evidence that caffeine (20–40 mg/kg) could normalize hepatic lipid content and blood transaminases. Moreover, caffeine was able to reduce the hepatic ATP/ADP ratio in a dose-dependent manner, indicating a decrease in energy metabolism, and activation of the *cAMP/CREB/SIRT3/AMPK/ACC* pathway. Subsequently, the authors demonstrated that suppression of some components of the pathway in oleate-treated HepG2 cells counteracted the caffeine effects. In summary, the study indicated that caffeine could ameliorate liver steatosis suppressing fatty acid synthesis and promoting β-oxidation. The observations also revealed that sirtuin-3 (*SIRT3*) was a key player in orchestrating caffeine effects. The observed protection involved the interleukin-6/signal transducer and activator of transcription 3 (*IL6/STAT3*) pathway and liver–muscle interorgan crosstalk. The experimental findings were also verified using in vitro cultures of myotubes and hepatocytes; and in a hepatocyte-specific *STAT3* knockout mouse model, demonstrating that the *IL6/STAT3* pathway was vital to the hepatoprotective effects of caffeine in NAFLD [28,29].

Aside from caffeine itself, the development of synthetic derivatives paved the way to future drug development. A new series of caffeine-8-(2-thio)-propanoic hydrazide–hydrazone derivatives was recently shown to exhibit pro-oxidant effects, and may be considered as promising structures for the design of future molecules with low hepatotoxicity [30].

### 2.2. Caffeic and Chicoric Acid 

Caffeic acid (IUPAC name: (E)-3-(3,4-dihydroxyphenyl)prop-2-enoic acid) is a phenolic acid (3,4-dihydroxycinnamic acid) present in several herbs and plants, including grapes, olives, spinach, asparagus, coffee, *Salvia miltiorrhiza*, and traditional Chinese herbs [31,32,33,34]. Among the properties at the biological level, it exhibits antioxidant potential, prevents DNA damage, has anticancer activity, inhibits low-density lipoprotein (LDL) lipid peroxidation, and is also able to reduce blood glucose levels. Using an in vivo mice model, Kim H.M. et al. [35] evidenced that CA prevented the hepatic steatosis induced by HFD and thus occurs through endoplasmic reticulum (ER) stress and autophagy regulation. In this model, CA reduced liver weight and hepatic triacylglycerol content by decreasing the expression of genes involved in fatty acid synthesis (such as sterol regulatory element-binding protein 1c or *SREBP-1c* and fatty acid synthase gene or *FAsN*), those engaged in β-oxidation, and ER stress markers (eukaryotic translation initiation factor 2a or *eIF2a*, activating transcription factor 4 or *ATF4*, and CCAAT-enhancer-binding protein homologous protein or *CHOP*). On the contrary, protein levels of autophagy markers (light chain 3 or *LC3*, autophagy related 7 or *ATG7*, and autophagy related 5 or *ATG5*) were significantly elevated in CA-fed obese mice. These results were also confirmed in vitro, using a palmitate-treated AML12 hepatocyte cell line exposed to CA [35]. 

Interestingly, Mu H-N. et al. [36] demonstrated in a similar mouse model that CA was able to revert the imbalance in the gut microbiota generated by the HFD. In addition, in this case, dietary supplementation with CA mitigated the body weight gain induced by HFD, with a reduction in liver weight and fat droplets. Moreover, they also observed the attenuation of hyperlipidemia and glycemia, as well as restored levels of ALT, in serum. At the biomolecular level, CA attenuated *SREBP1*, *FAS*, acetyl-CoA carboxylase (*ACC*), and stearoyl-CoA desaturase-1 (*SCD1*) expression, which are proteins involved in the de novo lipogenesis and generally upregulated in NAFLD. 

Recent experimental studies in animals have drawn attention to the role of LPS from the gut microbiota in favoring NAFLD occurrence. In agreement with the previous findings, the authors demonstrated that CA attenuated the increase in serum levels of lipopolysaccharide (LPS), tumor necrosis alpha (TNF-α), and interleukin 6 (IL-6), and several markers of hepatic inflammation (toll-like receptor 4 or *TLR4* expression, activation of phosphorylated *NF-κB* p65, and MPO-positive cell infiltration). Moreover, CA supplementation reduced the relative abundance of several bacteria that were proliferating under the HFD. In this context, the authors evidenced that CA was beneficial to balance gut LPS-related microbiota dysbiosis, preventing low-grade chronic inflammation in NAFLD [36].

Complementary information from in vitro studies using oleic-treated HepG2 cells indicated that CA’s beneficial effects on fatty acid accumulation depended on the upregulation of the AMP-activated protein kinase (*AMPK*) pathway [33]. A reduction in the expression of downstream target genes involved in lipogenesis (*ACC*, *SREBP-1*, *FAS*, glycerol-3-phosphate acyltransferase or *GPAT*, and 3-hydroxy-3-methylglutaryl-CoA reductase or *HMGCR*) and lipolysis (carnitine palmitoyltransferase 1 or *CPT-1*, peroxisome proliferator-activated receptor alpha or *PPAR-a*, and fatty acid binding protein 1 or *FABP-1*) was observed after CA exposure. Moreover, in HepG2 cells, Rafie H. et al. [37] demonstrated that CA and other polyphenols protected by more than 50% against ROS generation from oleic acid treatment by suppressing *TNF-α* expression and the induction of mitochondrial biogenesis. Additionally, Vergani L. et al. [38] demonstrated that CA and other polyphenolic compounds ameliorated lipid accumulation and lipid-dependent oxidative imbalance in both hepatic and endothelial cells, showing the potential of these compounds in nutraceutical formulations for tuning down NAFLD and atherosclerosis.

Chicoric acid (IUPAC name: (2R,3R)-2,3-bis[[(E)-3-(3,4-dihydroxyphenyl)prop-2-enoyl]oxy]butanedioic acid), a natural phenolic compound isolated from chicory (*Cichorium intybus*), is also present in other plants such as *Crepidistrum denticulatum* and *Echinacea purpurea* [39]. It exhibited many pharmacological properties, including anti-inflammatory [40], antioxidant [41], and antiviral [42] activities. Moreover, several reports indicated that ChiA may have beneficial effects in type 2 diabetes, hyperglycemia, obesity, and liver injury [43,44,45]. In this context, Kim M. et al. [45] investigated the capacity of *C. denticulatum* extract and ChiA to mitigate NASH in an methionine-choline-deficient (MCD) mouse model. Their observations demonstrated that the administration of ChiA or *C. denticulatum* extracts to MCD mice improved liver histology, reducing hepatic lipid contents and transaminases serum levels. Moreover, at the molecular level, a downregulation of crucial players in lipogenesis (*SREBP-1c*, *DGAT1*, *FAS*, and *SCD-1*), oxidative stress (*Nrf2*, *SOD1*, and catalase), inflammation (*NF-κB*, *TNF-α*, *IL-1b*, *IL-6*, and monocyte chemoattractant protein-1 or *MCP-1*), and fibrosis (alpha-smooth muscle actin or *α-SMA*, collagen type 1 A1 or *COL1A1*, collagen type 3 A1 or *COL3A1*, TIMP metallopeptidase inhibitor 1 or *TIMP-1*, and transforming growth factor beta or *TGFβ*) were found. These observations were replicated in vitro using two different cell models (HepG2 and AML-12). Cells were exposed to MCD medium and incubated with ChiA for 24 h, obtaining a downregulation of steatotic, proinflammatory, and profibrotic markers through the modulation of the *AMPK* pathway. Similar evidence was obtained by Ding X. et al. [46] both in vivo (HFD-fed mice) and in vitro (palmitic-acid-treated HepG2 cell model). ChiA treatment reversed HFD-induced oxidative stress and inflammation both systematically and locally in the liver, determined by quantifying MDA and SOD in serum and ROS in situ. In the amelioration of NAFLD, the activation of the *AMPK/Nrf2/NF-κB* signaling pathway and, interestingly, a modulation of the microbiota toward a healthier microbial profile (an increase in the Firmicutes-to-Bacteroidetes ratio) was also evidenced. Additional studies using in vitro models evidenced the synergistic benefits against NAFLD by combining ChiA with omega-3 fatty acids, achieving a tune regulation of *AMPK*-mediated *PPARα/UCP-2 SREBP-1/FAS* pathways [47]. It is also worth mentioning that Guo et al. [48] explored, in FFA-treated HepG2 cells, the role of circadian rhythm signaling during ChiA protection against fatty acid accumulation. The authors revealed that ChiA was a natural circadian clock modulator, regulating fatty acid anabolic and catabolic pathways in a BMAL1-dependent manner.

**Table 1 ijms-23-02764-t001:** Effects of coffee and its components on NAFLD.

Parameter	Compound	Model		Results	Ref.
		In Vitro	In Vivo		
**Steatosis**	Caffeine, green coffee extracts (GCE)		Female Sprague Dawley rats (HFD) 4.2–5.8 mg/kg/day	Neither caffeine nor GCE alleviated hepatic steatosis, but GCE-treated rats showed lower hepatic triglyceride levels	[49]
	Caffeine, chlorogenic acid		100 subjects with T2DM and NAFLD 200 mg caffeine with/without chlorogenic acid/day	Liver steatosis was not attenuated by caffeine or chlorogenic acid	[50]
	Coffee		2819 subjects with NAFLD or ALFD categorized consumption 0, 1, 2, and ≥3 cup/day	Coffee intake was not associated with any lower odds of hepatic steatosis	[51]
	Caffeine		Zebrafish in HFD 1–8% caffeine	Caffeine suppressed diet-induced hepatic steatosis by downregulation of genes associated with lipogenesis, ER stress, and inflammatory response	[52]
	Caffeine	HepG2 cells 2 mM	Male C57Bl/6 mice with HFD 10 and 20 mg/kg	Caffeine ameliorated hepatic steatosis by suppressing fatty acid synthesis and promoting β-oxidation	[52]
	Colombian coffee extracts		40 male Wistar rats (8–9 weeks old 30–70 mg/kg caffeine/day	Coffee extract attenuated diet-induced changes in structure and function of the liver and heart without changing the abdominal fat deposition	[53]
	Coffee		1452 subjects Caffeinated beverage consumption	No association between caffeine consumption and either the prevalence of fatty liver or serum ALT concentrations	[54]
	Caffeic acid	HepG2 cells 0–200 µM		Caffeic acid reduced lipid accumulation and increased *AMPK* phosphorylation, which reduced the expression of the genes involved in hepatic lipogenesis and increased those related to hepatic lipolysis	[33]
	Caffeic acid	AML12 cells 0–200 µM	Mice with HFD 50 mg/kg/day	Caffeic acid ameliorated hepatic steatosis, increasing autophagy and reducing ER stress	[45]
**Oxidative stress**	Caffeine		Male Wistar rats 20–30 mg/kg/day	Caffeine improved HFD-induced hepatic injury, suppressing inflammatory response, oxidative stress, and regulating lipogenesis and β-oxidation	[55]
	Caffeic acid	HepG2 cells 1, 5, and 10 µM		Polyphenols decreased ROS generation by oleic acid treatment, increasing the expression of markers of mitochondrial respiratory complex subunits and mitochondrial biogenesis	[37]
	Caffeic acid, other phenolic compounds	FaO cells 25 µM/24 h		Polyphenols ameliorated fatty acid accumulation and endothelial and hepatic lipid-dependent oxidative imbalance	[38]
	Chicoric acid	HepG2 cells 50–200 µM/24 h		Chicoric acid enhanced *Akt/GSK3b* signaling pathways and modulated the expression of downstream genes related to lipid metabolism in a BMAL1-dependent manner	[48]
**Inflammation**	Caffeine	Hepa 1-6, C2C12, and 3T3L1 cells 0.5 mg/mL	Male C57Bl/6 HFD	Caffeine ameliorated NAFLD via crosstalk between IL-6 production in muscle and liver *STAT3* activation	[28]
	Caffeic acid		Male C57Bl/6 HFD 0.08–0.16% caffeic acid supplementation HFD	Caffeic acid reverted the imbalance in the gut microbiota and related LPS-mediated inflammation, contributing to normalizing the dysregulation expression of lipid-metabolism-related genes	[36]
	Chicoric acid	HepG2 cells 10–20 µM/24 h	Male C57Bl/6 HFD 15–30 mg/kg/day	Chicoric acid modified gut microbiota toward a healthier microbial profile, ameliorating oxidative stress and inflammation via the *AMPK/Nrf2/NF-κB* signaling pathway	[46]
**Fibrosis**	Caffeine		195 severely obese subjects 0–5 g/wk total caffeine intake	Regular coffee consumption was an independent protective factor for liver fibrosis	[20]
	Caffeine		306 NAFLD subjects 0–822 (averaged 288 mg/day) mg/day total caffeine	Coffee consumption was associated with a significant reduction in the risk of fibrosis among NASH patients	[56]
	Caffeine, chlorogenic acid		Male TSOD mice spontaneous development of metabolic syndrome and NASH with liver tumors. 0.25 mg/caffeine day orally, 1.5 mg chlorogenic acid	Coffee consumption was associated with the prevention of metabolic syndrome; antifibrotic effects appeared to be due to the polyphenols rather than the caffeine	[57]
	Chicoric acid	HepG2 and AML12 cells 20 or 40 µM/24 h	Male C57BL/6 MCD diet 10–30 mg/kg/day	Chicoric acid reduced apoptosis, expression of lipogenesis-related genes, and fibrosis both in vivo and in vitro.	[45]

## 3. Tormentic Acid

Tormentic acid (TA) is a compound classified as a pentacyclic triterpene that is widely distributed in various plants and exhibits many pharmacological activities. It is also known as 2α,3β,19α-trihydroxyurs-2-en-28-oic acid (IUPAC name: (1R,2R,4aS,6aR,6aS,6bR,8aR,10R,11R,12aR,14bS)-1,10,11-trihydroxy-1,2,6a,6b,9,9,12a-heptamethyl-2,3,4,5,6,6a,7,8,8a,10,11,12,13,14b-tetradecahydropicene-4a-carboxylic acid), and the TA skeleton is composed of six isoprene units (C5) [58,59]. TA has been isolated in various plant foods such as strawberry fruit, olive, and the leaves of *Perilla frutescens*, *Eriobotrya japonica* Lindl, and *Potentilla tormentilla*, *Potentilla chinensis*, and *Sarcopoterium spinosum*. The majority of the species containing an abundant TA source belong to the *Rosaceae* family, mostly in the leaves and whole herbs [58,60]. This compound was found to possess various pharmacological properties, including hepatoprotective effects. 

TA was investigated in both in vitro and in vivo assays. Although relatively few studies have examined the effect of TA in NAFLD directly, substantial evidence was still found in the literature regarding its effects in NAFLD-related pathologies such as steatosis, oxidative stress, inflammation, and fibrosis. Table 2 summarizes the available data on TA activities and the mechanisms of its action.

*Rosa rugosa* roots containing TA were found to improve HFD-induced hyperlipidemia in rats via the activation of antioxidative mechanisms. HFD rats fed with TA lowered the HDL-, LDL-, and total cholesterol (TC) levels toward the values of the control group. Moreover, treatment of rats with TA increased antioxidative enzyme (SOD, glutathione peroxidase, and catalase) activities in hepatic tissues, suggesting that the compound prevented the loss of hepatic antioxidative activity produced by a high-fat diet [61].

Another study investigated the antihyperlipidemic and antihyperglycemic effects of TA derived from *Eriobotrya japonica* on HFD-fed mice. Treatment with TA significantly reduced the body weight gain, weights of white adipose tissue (WAT) (epididymal, perirenal, mesenteric WAT, visceral fat), and hepatic triacylglycerol content as compared to controls. Moreover, TA exhibited a hypolipidemic effect in HF-fed mice by decreasing gene expressions of fatty acid synthesis enzymes, including acyl-coenzyme A: diacylglycerol acyltransferase (*DGAT*) 2, which catalyzes the final step in the synthesis of triglycerides (TGs) [62]. 

Using the same mouse model, TA reduced visceral fat mass and hepatic triacylglycerol contents after HF diet. Moreover, TA significantly decreased both the area of adipocytes and ballooning degeneration of hepatocytes. The antihyperlipidemic effect was attributed to downregulation of the hepatic *SREBP-1c* and apolipoprotein C-III (*apo C-III*), and an increased *PPAR*-α expression [63].

In vitro and in vivo investigations have shown that TA has antidiabetic and normoglycemic properties. The methanolic extracts of *Potentilla fulgens* containing several triterpenes, including TA, were found to exhibit inhibitory activity against α-glucosidase. The intestinal α-glucosidase catalyzed the final step to release absorbable carbohydrates by hydrolyzing complex polysaccharides into oligosaccharides. Studies suggested that inhibition of this enzyme prevented a meal-induced increase in blood glucose levels [64]. The inhibition of protein tyrosine phosphatase 1B (*PTP1B)* is another proposed mechanism of TA’s antidiabetic effect. *PTP1B* enhances insulin sensitivity of the cells by negatively regulating tyrosine phosphorylation-dependent signals in various tissues, including insulin signaling [65]. Tormentic acid isolated from the plant *Eriobotrya japonica* Lindl effectively lowered blood glucose levels and triglyceride levels in HF-fed mice. Moreover, it was also observed that TA lowered visceral fat mass along with a reduction in free fatty acids and improved insulin resistance. The mechanism of reducing hyperglycemia acts mainly via increased skeletal muscular GLUT4 proteins that elevate glucose uptake but suppresses hepatic glucose production (downregulation of *PEPCK* and *G6 Pase*). The increased GLUT4 contents were shown to be mediated by enhanced *Akt* and *AMPK* phosphorylation both in skeletal muscle and in the liver, improving insulin sensitivity. TA increased hepatic fatty acid oxidation (*PPARα*) but suppressed lipogenic enzyme expression (including *SREBP-1c* and *FAS*), thus contributing to lowering triglyceride levels [62,63].

**Table 2 ijms-23-02764-t002:** Available data on tormentic acid activities in NAFLD-related pathologies.

Parameter	Model	Results	Ref.
Steatosis, Lipidemia	In vivo: HFD-fed rats	Inhibition of hyperlipidemia via the activation of the antioxidative mechanisms	[61]
	In vivo: HFD-fed mice	Reduction in body and adipose tissue weights Decreased expression of enzymes involved in fatty acid synthesis	[62]
	In vivo: HFD-fed mice	Reduced visceral fat mass and hepatic triacylglycerol contents Downregulation of *SREBP-1c* and *apo C-III*, and upregulation of *PPAR-α*	[63]
Glucose Homeostasis	In vitro: enzymatic assay	Inhibition of alpha-glucosidase activity	[64]
	In vitro: enzymatic assay	Inhibition of protein tyrosine *PTP1B* activity	[66]
	In vivo: HFD-fed mice	Decreased levels of blood glucose, insulin, leptin, and HOMA-IR index, and attenuated insulin resistance	[62]
Oxidative Stress	In vitro: rat vascular smooth muscle cells (RVSMCs)	Decreased ROS generation and downregulated the expression of iNOS and NADPH oxidase Prevented phosphorylation of *NF-κB* subunit p65 and degradation of the *NF-κB* inhibitor α (IκBα)	[67]
Inflammation	In vitro: rat vascular smooth muscle cells (RVSMCs)	Decreased levels of TNF-α, IL-6, and IL-1β Prevented phosphorylation of *NF-κB* subunit p65 and degradation of the *NF-κB* inhibitor α (*IκBα*)	[67]
	In vivo: acetaminophen-induced liver damage in mice	Inhibition of iNOS and COX-retention of enzymes (essential for the antioxidative properties of the liver): SOD, GPx, CAT Inhibition of *NF-κB* activation and inhibition of the activation of MAPKs	[68]
	In vitro: LPS-stimulated human gingival fibroblasts (HGFs)	Decreased expression of *IL-6* and *IL-8* Inhibited LPS-induced *TLR4* expression; *NF-κB* activation; *IκBα* degradation; and phosphorylation of ERK, JNK, and P38	[69]
	In vitro: LPS-induced inflammation in BV2 microglial cells	Inhibition of TNF-α and IL-1β Activation of *LXRα* and inhibition of *NF-κB* activation	[70]
	In vivo: acetaminophen-induced liver damage in mice	Reduction in TNF-α, IL-1β, and IL-6 Inhibition of *NF-κB* activation and inhibition of the activation of MAPKs	[68]
Fibrosis	In vitro: activated hepatic stellate cells	Decreased the expression of collagen type I and III Prevented excessive deposition of ECM	[71]

Aside from HFD-fed mice, TA was also evaluated in streptozoin diabetic rats. TA isolated from *Poterium ancistroides* improved the glucose tolerance test by increasing the insulin secretory response to glucose. However, no change in insulin and glucose levels was observed. The effect was similar to that of glibenclamide, suggesting that TA may act by increasing insulin secretion from the islets of Langerhans [72,73].

Regarding the antioxidant properties of TA, a study on hydrogen peroxide (H_2_O_2_)-induced oxidative stress in rat vascular smooth muscle cells (RVMCs) showed that TA exhibited antioxidant activities. TA was able to reduce reactive oxygen species (ROS) generation in RVSMCs exposed to H_2_O_2_. The study also demonstrated that TA downregulated the expression of inducible nitric oxide synthase (iNOS) and NADPH oxidase (NOX) through inhibition of the *NF-κB* signaling pathway [67]. In another study, the antioxidant properties of TA on acetaminophen (APAP)-induced liver damage were investigated in mice. TA was able to attenuate the APAP-induced production of nitric oxide (NO) and ROS. Furthermore, protein analysis revealed that TA inhibited iNOS, cyclooxygenase-2 (COX-2), and the activation of NF-κB and mitogen-activated protein kinases (MAPKs) [68]. 

In H_2_O_2_-induced inflammation in rat vascular smooth muscle cells (RVSMCs), TA significantly decreased the production of TNF-α, IL-6, and IL-1β. Furthermore, TA pretreatment inhibited H_2_O_2_-induced phosphorylation of the *NF-κB* subunit p65 and degradation of the *NF-κB* inhibitor (*IκBα*) in RVSMCs. TA was therefore suggested to inhibit H_2_O_2_-induced inflammation in RVSMCs through suppression of the *NF-κB* signaling pathway [67]. Similarly, a study investigated the anti-inflammatory effects of TA on lipopolysaccharide (LPS)-stimulated human gingival fibroblasts (HGFs). The results showed that TA significantly inhibited the LPS-induced IL-6 and interleukin 8 (IL-8) production in a dose-dependent manner. Furthermore, TA inhibited LPS-induced TLR4 expression; *NF-κB* activation; *NF-κB* inhibitor (*IκBα*) degradation; and phosphorylation of extracellular signal-regulated kinase (*ERK*), c-Jun N-terminal kinase (*JNK*), and *p38* [69]. Similar results were obtained in LPS-induced neuroinflammation in BV2 microglia cells. Treatment of TA downregulated the expression of *TNF-α* and *IL-1β* by inhibiting *NF-κB* and activating liver X receptor alpha (*LXRα*) receptors [70]. 

The above results were supported in an in vivo study of acetaminophen (APAP)-induced liver damage. TA significantly decreased the serum IL-1β, IL-6, and TNF-α levels in mice. As with the in vitro studies, the anti-inflammatory effect was attributed to the inhibition of *NF-κB* and mitogen-activated protein (MAP) kinase activities [68].

## 4. Verbascoside

Verbascoside, also known as acteoside, is a phenylethanoid glycoside with the IUPAC name 6-[2-(3,4-dihydroxyphenyl)ethoxy]-5-hydroxy-2-(hydroxymethyl)-4-(3,4,5-trihydroxy-6-methyloxan-2-yl)oxyoxan-3-yl] 3-(3,4-dihydroxyphenyl)prop-2-enoate [74]. Verbascoside can be found in more than 200 plant species, including *Plantago* and *Lippia* species [61]. Accumulating evidence has shown that verbascoside can exert various pharmacological activities, such as antioxidant, antimicrobial, anti-inflammatory, neuroprotective, anticancer, and hepatoprotective effects [75]. As with TA, only a few studies examined the effect of verbascoside in NAFLD directly. However, there is substantial evidence that showed the effects of verbascoside in NAFLD/NASH-related pathologies such as steatosis, oxidative stress, inflammation, and fibrosis. Table 3 summarizes the available data on verbascoside activities and the mechanisms of its action.

In a HFD rat model, a reduction in body weight was observed between weeks 12 and 18 in the group cotreated with verbascoside as compared to the placebo group. Moreover, verbascoside ameliorated the serum lipid profile by lowering TC, TGs, and LDL. The data also suggested that the significant reduction in lipid levels by verbascoside might be attributed to the regulation of the *AMPK* and *mTOR* pathways [76].

In a double-blind, placebo-controlled, and randomized trial in 56 obese/overweight subjects, the effects of a dietary supplement containing 500 mg of a combination of polyphenolic extracts from *Lippia citriodora* L. and *Hibiscus sabdariffa* L. (LC-HS) were evaluated. Among the phenylpropanoids, verbascoside represented the major compound present in the supplement. After two months of the trial, the consumption of the LC-HS polyphenols showed significant improvements in body weight, the abdominal circumference of overweight subjects, and body fat % as compared to controls [77].

Substantial epidemiological evidence indicated that a diet rich in polyphenols, particularly verbascoside, protected against developing type 2 diabetes. In an in vitro study utilizing mouse and human pancreatic β-cells, verbascoside protected the cells from the ER stress. Mechanistic studies revealed that verbascoside mitigated the activation of the protein kinase RNA-like endoplasmic reticulum kinase (*PERK*) branch of the unfolded protein response, thereby encouraging mitochondrial dynamics. As a result, pancreatic β-cells showed improved viability, mitochondrial activity, and insulin content [78].

The antidiabetic effects of verbascoside were evaluated in streptozotocin–nicotinamide (STZ–NA)-induced type 2 diabetic rats. Results revealed that verbascoside-treated rats had lower blood glucose levels, glycosylated hemoglobin, total cholesterol, triglycerides, and increased serum insulin compared to control diabetic rats. These effects were comparable to those caused by the standard antidiabetic drug pioglitazone [79].

Verbascoside from *Acanthus mollis* leaves showed antioxidant properties in both the HepG2 and SH-SY5Y cell lines. In vitro scavenging activities of verbascoside using 2,2-diphenyl-β-picrylhydrazyl (DPPH), hydroxyl, and superoxide assays were comparable to ascorbic acid as the reference substance [80]. The antioxidant effects of verbascoside were also evaluated in streptozotocin–nicotinamide (STZ–NA)-induced type 2 diabetic rats. Verbascoside-treated rats, in comparison to the diabetic control, demonstrated significantly reduced malondialdehyde, increased reduced glutathione liver contents, and attenuated pathological alterations in the liver, suggesting antioxidant properties. Furthermore, verbascoside scavenged the stable free radical 1,1-diphenyl-2-picrylhydrazyl in vitro [79].

Using an atherosclerotic high-fat-diet rat model, verbascoside was able to ameliorate the serum levels of inflammatory mediators. IL-1b, IL-6, high-sensitivity C-reactive protein or hs-CRP, and matrix metallopeptidase 9 or MMP 9 were decreased, and IL-10 was increased in the verbascoside and simvastatin-treated groups compared to the atherosclerosis group, possibly through regulating the expression of the *AMPK* and *mTOR* protein [77].

In the human prostate cancer cell lines Du-145 and PC-3, verbascoside was able to downregulate the expression level of alpha-smooth muscle actin by inhibiting the transforming growth factor (TGF-β)/Smad signaling pathway [81]. 

In addition, the effects of verbascoside in rats with renal fibrosis through unilateral ureteral obstruction were evaluated. Results showed that fibrosis-related proteins, including collagen type I (*COL-I*), α-smooth muscle actin (*α-SMA*), and tissue inhibitor of metalloproteinase 2 (*TIMP2*), were lowered by verbascoside. In addition, verbascoside was able to ameliorate macrophage infiltration and alleviate the degree of renal fibrosis histologically [82].

**Table 3 ijms-23-02764-t003:** Available data of verbascoside activities in NAFLD-related pathologies.

Parameter	Model	Results	Ref.
Steatosis, Lipidemia	In vivo: HFD-fed rats	Reduction in body weight Ameliorated serum lipid profile	[76]
	56 obese/overweight (2 months)	Improvements in body weight, abdominal circumference, and % body fat	[77]
Glucose homeostasis	In vitro: mouse and human pancreatic β-cells	Increased viability, mitochondrial function, and insulin content of pancreatic β-cells	[78]
	In vivo: streptozotocin–nicotinamide (STZ–NA)-induced type 2 diabetic rats	Lower levels of blood glucose, glycosylated hemoglobin, and increased serum insulin	[79]
Oxidative stress	In vitro: HepG2 and SH-SY5Y cell lines	Improved DPPH, OH, and O_2_ scavenging activities	[80]
	In vivo: streptozotocin–nicotinamide (STZ–NA)-induced type 2 diabetic rats	Reduction in MDA levels and restored GSH in livers of diabetic rats	[79]
Inflammation	In vivo: high-fat-fed rats	Reduction in serum inflammatory markers	[76]
Fibrosis	In vitro: Du-145 and PC-3 cell lines	Reduction in *α-SMA* expression	[81]
	In vivo: renal-fibrosis-induced rats	Reduction in *COL-I*, *α-SMA*, and *TIMP2* Decreased macrophage infiltration	[82]

## 5. Silymarin (*Silybum marianum*)

Silymarin (IUPAC name: 3,5,7-trihydroxy-2-[3-(4-hydroxy-3-methoxyphenyl)-2-(hydroxymethyl)-2,3-dihydro-1,4-benzodioxin-6-yl]-2,3-dihydrochromen-4-one), extracted from the plant seeds and fruits of *Silybum marianum* (commonly known as milk thistle), has been used as a medicinal herb since as early as the 4th century B.C. [83,84,85]. It has been widely employed in the treatment of various liver disorders due to its hepatoprotective properties such as anti-inflammatory [86,87,88,89], antiproliferative [90,91,92], immunomodulatory [93], and anticholesterolemic [94,95]. The free radical scavenging property is the most interesting aspect of silymarin as a therapeutic compound [83,84]. Free radical scavenging enzymes such as SOD, catalase, and glutathione peroxidase protect cells from oxidative stress. Both silymarin [96,97] and silibinin can increase the expression of SOD in lymphocytes of patients with chronic alcoholic liver disease. In line with these findings, studies performed in NAFLD animal models (summarized in Table 4) and in patients with liver disease indicate that silibinin exerts its antioxidant properties by increasing the levels of glutathione, glutathione peroxidase, and SOD [98,99]. Furthermore, it also was reported that silibinin decreased the production of ROS and the lipoperoxidation products, such as malondialdehyde (MDA), observed in NAFLD [98,100,101,102]. Interestingly, by reducing the oxidative damage, silibinin improves insulin sensitivity [100], thus reducing the glycemia and insulinemia and improving the homeostatic model assessment for insulin resistance (HOMA) index. From a mechanistic point of view, it has been proposed that the insulin receptor substrate-1 (*IRS-1*)/*PI3K/Akt* pathway seemed to be inversely correlated to the pathogenesis of NAFLD, since knockout of *IRS-1* [103] and recovery of the *PI3K/Akt* activity [104] protects the liver from NASH-induced injury. Recent data [98] have shown that silibinin is able to restore the levels of phosphorylated *IRS-1*, total *IRS-1*, *PI3K*, and phosphorylated Akt, which are inhibited in NAFLD. The increased ROS generation also leads to the production of proinflammatory cytokines through activation of the *MAPK* pathway [105]. The proinflammatory cytokine TNF-α, mainly produced by Kupffer cells, plays a pivotal role in several liver diseases, including alcoholic hepatitis and nonalcoholic steatohepatitis (NASH). TNF-α signaling is regulated by both *MAPK* and *NF-κB* [106] in a crosstalk considered important for the hepatocyte homeostasis [107]. Silibinin exerts an anti-inflammatory effect by decreasing the levels of some of the proinflammatory cytokines, such as IL-8 [108], IL-6 [98], MCP-1 [108], and TNF-α [109,110]. This effect is a consequence of the inhibition of *NF-κB* activation via the inhibition of *IKK-beta* and a decreased activity of *p50* and *p65*. Apoptosis has been recognized as a crucial event in many liver injuries, and the activation of procaspase-3 has been documented as a prominent pathological feature in NASH patients and NASH animal models [111,112]. It has been shown that exposure to silibinin is able to suppress the activation of procaspase-3 to caspase-3 [110]. Moreover this compound also was reported to inhibit TNF-induced *JNK* and *MEK* (a MAPK upstream kinase) activation in a dose-dependent manner (1–50 µM), thus also inhibiting TNF-induced apoptosis [113].

Silibinin showed antifibrogenic effects by inhibiting collagen type I biosynthesis in animals with secondary biliary fibrosis [114,115]. Furthermore, this compound was able to inhibit HSC activation [116] (in terms of *α-SMA* expression) and PDGF-induced proliferation and migration [108]. Data from the rat models of CCl_4_-induced liver damage showed an increased *ERK* activity in HSC, suggesting a key role in the proliferation and migration of this cell type [117]. Along this line was the intriguing evidence of an inhibition of the entire *ERK* cascade (*Raf*, *MEK*, and *ERK*) by silibinin (10–100 µM) [118]. 

In the pathogenesis of NAFLD, an augmented hepatic de novo lipogenesis plays a central role. Hepatic de novo lipogenesis is combined with dyslipidemia (increased plasma TG and decreased VLDL clearance), leading to ectopic and whole-body lipid deposition. Silibinin has been shown to reduce the thrombotic complications associated with this event by reducing the hepatic TG content [101], as well as serum TG and total cholesterol levels [119]. Indeed, histological data from several independent studies showed that silibinin promoted an improvement in liver steatosis, inflammation, and cell ballooning (summarized in Table 4). 

Despite the previously reported data, silymarin has been widely proposed in the treatment of NASH, but definitive data have not been provided so far. Due to its low bioavailability [120,121], some solubilizing compounds (phosphatidylcholine; β-cyclodextrin, and vitamin E) were added to the plant extracts to enhance intestinal absorption [122]. Although data obtained in animal models [123] are promising, the administration and the dosage are often difficult to be reproduced in humans. On the other hand, data from clinical trials using silymarin [109,124] are controversial, and the real efficacy has been questioned for years (reviewed elsewhere [125]). The main limitations are the lack of silymarin standardization among its various formulations and the still-undefined effective dosage [125]. This situation is even worse among the pediatric and juvenile population, in which the available information is scarce. Studies of dietary supplements in children have shown inconsistent effects to benefit children with NAFLD [126]. 

Finally, when it comes to the biosynthetic derivatives of silybin, most were found to be effective radical scavengers and lipid peroxidation inhibitors [127]. Although not in a NAFLD model, one particular interest was in 3-O-palmitoyl-silybin, a de novo synthetized compound that had stronger antioxidant and anti-lipoperoxidant protective effects than sylibin [128].

**Table 4 ijms-23-02764-t004:** Silymarin effects in different in vivo models of NAFLD.

Parameter	Model	Results	Ref.
Liver histology	Rats (8–9 weeks old) + HFD (100 mg/kg daily orally) for 12 weeks	Improved steatosis Reduced inflammatory foci	[129]
	Db/db mice (6 weeks old) + MCD 20 mg/kg daily IP (4 weeks)	Improved steatosis	[123]
	Db/db mice (8 weeks old) + MCD 20 mg/kg daily IP (4 weeks)	Improved steatosis Reduced lobular inflammation Decreased cell ballooning	[102]
	OLETF rats + MCD 0.5% w/w of diet orally (8 weeks)	Improvement of the NAS score Improvement of fibrosis (by reducing HSC activation)	[116]
	Male rats + MCD diet 1 g seed powder/kg daily gavages (3 weeks)	Improvement of steatosis, inflammation, and cell ballooning	[110]
	Gerbils + HFD 100 mg/kg daily by gastric intubation (8 weeks)	Improved steatosis	[130]
	Rats (4–6 weeks old) + HFD 25 mg/kg daily intragastric (6 weeks)	Decreased the fatty degeneration and the lobular inflammation	[119]
Glucose homeostasis	Rats (8–9 weeks old) + HFD 100 mg/kg orally (12 weeks)	Improved insulin sensitivity	[129]
	Rats (8 weeks old) high-fructose diet 100–300 mg/kg daily orally (3 weeks)	Decreased glycemia Decreased insulinemia Improved HOMA-IR	[101]
	Db/db mice (6 weeks old) + MCD 20 mg/kg daily IP (4 weeks)	Decreased glycemia Decreased insulinemia Improved HOMA-IR	[123]
	Gerbils + HFD 100 mg/kg daily by gastric intubation (8 weeks)	Decreased glycemia Decreased insulinemia	[130]
	Rats (4–6 weeks old) + HFD 25 mg/kg daily intragastric (6 weeks)	Improved HOMA-IR	[119]
Oxidative stress	Db/db mice (6 weeks old) + MCD 20 mg/kg daily IP (4 weeks)	Decreased lipoperoxidation Restored the GSH and nitrite/nitrate levels	[123]
	Db/db mice (8 weeks old) + MCD 20 mg/kg daily IP (4 weeks)	Decreased lipoperoxidation, TBARS, and ROS	[102]
	Rats (8 weeks old) high-fructose diet 100–300 mg/kg daily orally (3 weeks)	Decreased MDA and nitrite content	[101]
	Male rats + MCD diet 1 g seed powder/kg daily gavages (3 weeks)	Decreased MDA and improved GSH	[110]
	Gerbils + HFD 100 mg/kg daily by gastric intubation (8 weeks)	Decreased lipoperoxidation	[130]

## 6. Discussion

Despite the increasing incidence of NAFLD and the related worrisome future perspectives, to date, there is no pharmacological treatment to revert the onset or to avoid the progression from fatty liver to the more severe stages of the disease. Despite the scientific community’s efforts, altogether, the relatively benign and reversible characteristics of the early stages of this disorder, the multifactorial events associated with the worsening of liver functionality, and the limitations of the tools for an early diagnosis hamper the development of an effective drug. It has been well established that lifestyle modification, mainly through diet and an increase in physical activity, significantly improve many of the factors associated with NAFLD. Considering that NAFLD is tightly associated with obesity (in which sedentarism and an unhealthy diet represent the basis of this condition), obtaining successful results by this approach is not always possible, making this alternative inefficient. A promising option to overcome the reticence toward lifestyle changes would be the use of compounds naturally present in food with reported beneficial effects in counteracting the deleterious events associated with fat accumulation. 

Here, we reviewed the selected food derivatives with beneficial effects in NAFLD reported by preclinical and clinical studies. Although NAFLD is a multifactorial disease with complex mechanisms as proposed in the “multiple hit” model [7], the pathogenesis could be initially explained by hepatic fat accumulation or steatosis. There are three major mechanisms identified as the source of excessive fat accumulation into the liver: (a) increased visceral adipose tissue (AT) lipolysis; (b) de novo lipogenesis (DNL) by consumption of excess calories; and (c) insulin resistance [4,5,9]. Lipid accumulation predisposes the liver to lipotoxicity, which triggers inflammation, mitochondrial dysfunction, and oxidative stress, leading to steatohepatitis and/or hepatic fibrosis [5]. These critical pathological events are considered important therapeutic targets to delay the progression of NAFLD. Figure 1 illustrates the pathogenesis of NAFLD and the mechanism of action of the compounds included in this review.

Coffee is one of the most commonly consumed beverages worldwide, with epidemiological evidence showing that its consumption is protective against several diseases, including liver fibrosis, cirrhosis, chronic liver disease, and liver cancer [131]. Specifically, in the context of NAFLD, the evidence collected over time is quite convincing; several of its derivatives have demonstrated promising effects in improving many of the events associated with the disease (Table 1). Overall, caffeine and its derivatives caffeic acid and chicoric acid were able to counteract steatosis, inflammation, oxidative stress, and fibrosis, as shown by in vitro and in vivo studies. Of note, caffeic acid and chicoric acid were also able to modulate the gut microbiota toward a healthy microbial profile, thereby attenuating gut dysbiosis. Gut microbiota, along with other nonhepatic players, are important components of the complex “multiple hit” model.

However, after all these years of study, we still cannot establish an effective therapeutic scheme focused on coffee. This is the result of different aspects related to coffee. For instance, the different coffee species determine the chemical composition of coffee brews. These differences contribute to the characteristic flavor and quality of coffee beverages made from each species. Typically, green robusta seeds contain almost twice as much caffeine, more chlorogenic acids, and less trigonelline than arabica per weight unit. Moreover, the chemical composition varies greatly depending on the quality of the coffee and roasting degree [132,133]. Other variables are the modalities of coffee preparation (controversies remain regarding whether espresso or filtered coffee have the same beneficial effects on NAFLD) and the daily average of coffee consumption among studies. Additionally, most of the trials conducted so far have either been retrospective, observational, and/or cross-sectional point prevalence studies. Due to the lack of prospective data, the conclusions can be drawn only from the association between coffee drinking and liver health. The sum of all these variables resulted in a large heterogeneity among the studies, which further complicated the reaching of definitive conclusions. Thus, more interventional and prospective trials under standardized conditions are still needed to define an effective therapeutic guideline.

More recently, tormentic acid (TA) extract, isolated from various plant foods such as strawberry fruit, olive, and the leaves of different plants from the *Rosaceae* family, was found to possess various pharmacological properties, including hepatoprotective effects. To date, relatively few studies have examined the impact of TA in NAFLD directly; however, substantial evidence has been reported on its effects in NAFLD-related pathologies such as steatosis, oxidative stress, inflammation, and fibrosis. The data collected so far (summarized in Table 2) suggested that TA might have promising prospects in the future. Interestingly, TA reduced visceral fat mass, hepatic triacylglycerol contents, and serum triglycerides in an HF-diet mouse model (Figure 1). However, the lack of solid results in the context of NAFLD, with appropriate dosage and time of treatment, hamper its use in future clinical trials.

Another emerging compound analyzed herein is verbascoside, which can be found in more than 200 plant species. This compound can exert various pharmacological activities such as antioxidant, antimicrobial, anti-inflammatory, neuroprotective, anticancer, and hepatoprotective effects, which could be interesting in the context of NAFLD. The preclinical and clinical studies reported herein suggested that verbascoside exerted protective effects against NAFLD-related pathologies (Table 3 and Figure 1). The data collected provided a rationale for the possible use of this compound as a nutraceutical in disease prevention and treatment. Reliable clinical studies in the context of NAFLD are limited, and must be expanded together with pharmacodynamics and pharmacokinetic studies for future application of verbascoside in a clinical setting. 

On the other hand, silymarin, which is extracted from plant seeds and fruits of *Silybum marianum*, has been used as a medicinal herb from as early as the 4th century B.C. Silymarin has been widely proposed in the treatment of NASH, although definitive data have not been provided (reviewed elsewhere [125]). Data from clinical trials using silymarin are controversial, and the real efficacy has been questioned for years. The main limitations are the lack of silymarin standardization among its various formulations and the still-undefined effective dosage. Another limitation is its low bioavailability [120,121]; some solubilizing compounds (phosphatidycholine; β-cyclodextrin, vitamin E) were added to the plant extracts to enhance intestinal absorption [122]. An additional limitation of the use of silymarin is the low comparability among studies; for instance, different formulations of the drug produced different bioavailabilities and dispositions, even when the same dose was administered. Additionally, despite the promising data coming from animal models, the administration and the dosage are often difficult to reproduce in humans. A better definition of the clinical trial conditions is still necessary to assess the real efficacy of silymarin in NAFLD. 

## 7. Materials and Methods

For this review, we used the Pubmed and Google Scholar databases to search for relevant articles using the following mesh terms: “Coffee”, “Caffeine”, “Caffeic Acid”, “Chicoric Acid”, “Verbascoside”, “Tormentic Acid”, “Silybum Marianum”, “Silymarin”, “Silybinin”, “Silybin”, “NASH”, “NAFLD”, “Steatosis”, “oxidative stress”, “liver inflammation”; “liver fibrosis”, and “Diabetes”. If not differently stated in the respective section, we did not consider the specific time frame.

## 8. Conclusions

NAFLD is a chronic liver disease that includes many pathological aspects, including steatosis, oxidative stress, inflammation, and fibrosis. Moreover, the presence of multiple mechanisms and comorbidities associated with NAFLD has hampered the search for an effective drug for the disease. Natural compounds have been studied preclinically and clinically, providing evidence of their beneficial effects in liver diseases such as NAFLD. Although lifestyle modifications involving diet and exercise currently remain the first line of treatment for NAFLD, it appears that the compounds reviewed here could also potentially improve NAFLD treatment. Overall, the need for effective pharmacologic treatment for NAFLD, and most importantly, NASH, is evident, but we are still far from achieving this socioeconomic goal. The issue must be approached using a coordinated translational effort, which will provide reliable and effective tools to reduce the NAFLD pandemic. 

## Figures and Tables

**Figure 1 ijms-23-02764-f001:**
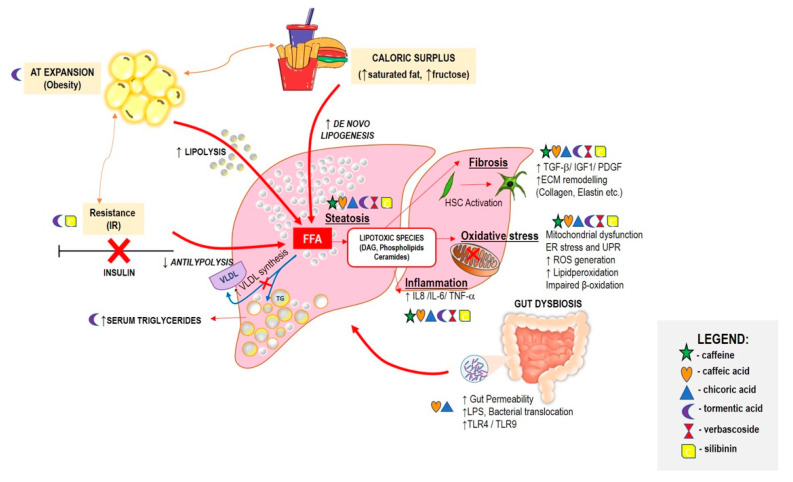
Pathophysiological mechanisms of NAFLD and some critical events counteracted by the compounds. Adipose tissue expansion, insulin resistance, and caloric surplus can lead to free fatty acid accumulation in the liver, which inhibits VLDL synthesis, and thus increases the TG intrahepatic pool. These events, along with impaired with β-oxidation, promote steatosis. Lipotoxic species can then cause oxidative stress, inflammation, and fibrosis. The activation of hepatic stellate cells marks the promotion of NASH fibrosis. Nonhepatic players can also contribute directly or indirectly to NASH progression. Changes in gut microbiota composition can yield toxic microbiota products, or even form a leaky gut to release LPS or bacteria, all of which could contribute to hepatic inflammation. The compounds’ beneficial effects on NAFLD can be attributed to counteracting these critical pathological events and other nonhepatic players. Abbreviations: DAG, diacylglycerol; ECM, extracellular matrix; FFA, free fatty acids; IL-6, interleukin-6; IL-8, interleukin-8; IR, insulin resistance; IGF-1, insulin-like growth factor 1; LPS, lipopolysaccharides; PDGF, platelet-derived growth factor; TLR, toll-like receptor; TGF-β, transforming growth factor beta; TNF-α, tumor necrosis—alpha; VLDL, very-low-density lipoprotein.

## Data Availability

The data presented in this study are openly available online in Pubmed and Google Scholar.

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
