# Peer review of "Natural Compounds for Counteracting Nonalcoholic Fatty Liver Disease (NAFLD): Advantages and Limitations of the Suggested Candidates"

_ijms, 2022, doi:10.3390/ijms23052764_

Round 1

Reviewer 1 Report

the review article titlled "Natural compounds for counteracting NAFLD/NASH: Advantages and limitations of the suggested candidates" describes an interestign research topic in which natural products are involved.

The manuscript is well written and organized, although some minor points need to be revised before publication:

-the introduction must be improved. It is too short.

-the chemical strctures of the described compounds should be inserted in the main text.

-the semisynthetic derivatives of these boioactives, if described for the same target must be introduced in a paragraph.

- a comprehensive cartoon of the mechanism of the target should enrich the review value.

Author Response

RESPONSE TO REVIEWER 1

The review article titlled "Natural compounds for counteracting NAFLD/NASH: Advantages and limitations of the suggested candidates" describes an interesting research topic in which natural products are involved.
The manuscript is well written and organized, although some minor points need to be revised before publication:

R: Thank you for your comments and your time, please find below the comments/actions to your observations.

Comment #1:
-the introduction must be improved. It is too short.
Additional details have been added in the introduction. 

Comment #2:
-the chemical structures of the described compounds should be inserted in the main text.
IUPAC names of the compounds were included as described in Pubchem. 

Comment #3:
-the semisynthetic derivatives of these bioactives, if described for the same target must be introduced in a paragraph.
Although not directly on NAFLD, the beneficial effects of some semisynthetic derivatives were described in the revised manuscript. 

Comment #4:
- a comprehensive cartoon of the mechanism of the target should enrich the review value.
A figure was added showing the pathophysiology of NAFLD and the events counteracted by the compounds.

Reviewer 2 Report

The review by Salvoza et al. critically addresses and discusses the validity of natural compounds used in experimental and clinical studies as treatments of NAFLD / NASH. The manuscript has the merit of carefully addressing the natural compounds examined and critically reviewing the related literature.The tables help the reader to interpret the different studies proposed. Overall, the manuscript is useful e properly focused.

Only minor changes could be suggested:

The discussion is somewhat redundant compared to the rest of the text and the conclusions are too simplified. Paragraphs can probably be better balanced. There are several text errors and typos.

Author Response

RESPONSE TO REVIEWER 2

The review by Salvoza et al. critically addresses and discusses the validity of natural compounds used in experimental and clinical studies as treatments of NAFLD / NASH. The manuscript has the merit of carefully addressing the natural compounds examined and critically reviewing the related literature.The tables help the reader to interpret the different studies proposed. Overall, the manuscript is useful e properly focused.
Only minor changes could be suggested:
R: Thank you for your comments and your time, please find below the comments/actions to your observations.

Comment #1:
The discussion is somewhat redundant compared to the rest of the text and the conclusions are too simplified. Paragraphs can probably be better balanced. There are several text errors and typos.

R: Misspellings and typos were corrected. Discussion was revised. A figure was added showing the pathophysiology of NAFLD and the events counteracted by the compounds. Conclusion was made comprehensive

Reviewer 3 Report

The review “Natural compounds for counteracting NAFLD/NASH: Advantages and limitations of the suggested candidates”  is prepared professionally.

It includes a well-crafted abstract and an exhaustive introduction that justifies the research undertaken.

The introduction points to the deficiencies in the literature on the subject.

The aim is clearly defined.

Modern analytical methods were used in the research.

The discussion of the results is well prepared.

The conclusions are well-defined.

The illustrative material is appropriate.

In my opinion, the manuscript after corrections, will be suitable for publication in a journal.

Detailed comments:

Title: Please do not abbreviation like NAFLD/NASH in title

Abstract - The abstract is not enough in my opinion. Must be extend

In addition avoid to use abbreviation when use  first time. 

Introduction: is short and should be increased to double

Below sentence needs several references (updated references)

Several natural compounds have been shown to exert beneficial effect(s) in the cellular mechanisms involved in the onset and progression of different diseases. These pieces of evidence represent a promising strategy for NALFD, whose pathogenesis is multifactorial involving steatosis, inflammation, oxidative stress, fibrosis, apoptosis, among others. Despite the reported results on experimental models, the translation to clinical setting 42 has been disappointing because of the variations in the effective dosage, the bioavailability, the duration of the treatment, the differences in the purity of the compound, and the 44 lack of standardization. 

References not given in journal style. For example [7][8] must be [7,8].

The most of the section of papers needs references

Table prepared in general careless. For example there is no title in Table 1.

The title at the bottom wrongly given

Author Response

RESPONSE TO REVIEWER 3

The review “Natural compounds for counteracting NAFLD/NASH: Advantages and limitations of the suggested candidates” is prepared professionally.
It includes a well-crafted abstract and an exhaustive introduction that justifies the research undertaken.
The introduction points to the deficiencies in the literature on the subject.
The aim is clearly defined.
Modern analytical methods were used in the research.
The discussion of the results is well prepared.
The conclusions are well-defined.
The illustrative material is appropriate.
In my opinion, the manuscript after corrections, will be suitable for publication in a journal.

R: Thank you for your comments and your time, please find below the comments/actions to your observations.

Detailed comments:

Comment #1:
Title: Please do not abbreviation like NAFLD/NASH in title
The title was edited accordingly

Comment #2:
Abstract - The abstract is not enough in my opinion. Must be extend
In addition avoid to use abbreviation when use  first time. 
R: All points have been addressed. Details in the abstract were added. Usage of abbreviation was done properly.

Comment #3:
Introduction: is short and should be increased to double
R: Additional details have been added in the introduction. 

Comment #4:
Below sentence needs several references (updated references)
“Several natural compounds have been shown to exert beneficial effect(s) in the cellular mechanisms involved in the onset and progression of different diseases. These pieces of evidence represent a promising strategy for NALFD, whose pathogenesis is multifactorial involving steatosis, inflammation, oxidative stress, fibrosis, apoptosis, among others. Despite the reported results on experimental models, the translation to clinical setting has been disappointing because of the variations in the effective dosage, the bioavailability, the duration of the treatment, the differences in the purity of the compound, and the lack of standardization”
R: References were added

Comment #5:
References not given in journal style. For example [7][8] must be [7,8].
The most of the section of papers needs references
R: Lacking references were added. Reference style was updated according to MDPI format

Comment #6:
Table prepared in general careless. For example there is no title in Table 1.
The title at the bottom wrongly given
R: Table format and table captions were improved accordingly.

Round 2

Reviewer 3 Report

Dear Editor,

The authors made all necessary changes and additions and I believe that the paper now is ready for publication.